# Integrative Genomics and Bioactivity-Guided Isolation of Novel Antimicrobial Compounds from *Streptomyces* sp. KN37 in Agricultural Applications

**DOI:** 10.3390/molecules29092040

**Published:** 2024-04-28

**Authors:** Jing Zhao, Qinghua Li, Muhammad Zeeshan, Guoqiang Zhang, Chunjuan Wang, Xiaoqiang Han, Desong Yang

**Affiliations:** The Key Laboratory of Oasis Agricultural Pest Management and Plant Protection Utilization, College of Agriculture, Shihezi University, Shihezi 832003, China; 20212012063@stu.shzu.edu.cn (J.Z.); 20201012214@stu.shzu.edu.cn (Q.L.); maharzeeshan218@gmail.com (M.Z.); hanshz@shzu.edu.cn (X.H.); yds_agr@shzu.edu.cn (D.Y.)

**Keywords:** *Streptomyces* sp. KN37, secondary metabolites, whole-genome sequencing, 4-(Diethylamino)salicylaldehyde, 4-Nitrosodiphenylamine

## Abstract

Actinomycetes have long been recognized as an important source of antibacterial natural products. In recent years, actinomycetes in extreme environments have become one of the main research directions. *Streptomyces* sp. KN37 was isolated from the cold region of Kanas in Xinjiang. It demonstrated potent antimicrobial activity, but the primary active compounds remained unclear. Therefore, we aimed to combine genomics with traditional isolation methods to obtain bioactive compounds from the strain KN37. Whole-genome sequencing and KEGG enrichment analysis indicated that KN37 possesses the potential for synthesizing secondary metabolites, and 41 biosynthetic gene clusters were predicted, some of which showed high similarity to known gene clusters responsible for the biosynthesis of antimicrobial antibiotics. The traditional isolation methods and activity-guided fractionation were employed to isolate and purify seven compounds with strong bioactivity from the fermentation broth of the strain KN37. These compounds were identified as 4-(Diethylamino)salicylaldehyde (1), 4-Nitrosodiphenylamine (2), N-(2,4-Dimethylphenyl)formamide (3), 4-Nitrocatechol (4), Methylsuccinic acid (5), Phenyllactic acid (6) and 5,6-Dimethylbenzimidazole (7). Moreover, 4-(Diethylamino)salicylaldehyde exhibited the most potent inhibitory effect against *Rhizoctonia solani*, with an EC_50_ value of 14.487 mg/L, while 4-Nitrosodiphenylamine showed great antibacterial activity against *Erwinia amylovora*, with an EC_50_ value of 5.715 mg/L. This study successfully isolated several highly active antimicrobial compounds from the metabolites of the strain KN37, which could contribute as scaffolds for subsequent chemical synthesis. On the other hand, the newly predicted antibiotic-like substances have not yet been isolated, but they still hold significant research value. They are instructive in the study of active natural product biosynthetic pathways, activation of silent gene clusters, and engineering bacteria construction.

## 1. Introduction

Plant diseases are a major cause of significant economic losses in plant production. With the widespread use of pesticides, issues such as the increased resistance of pathogens, escalating prevalence of plant diseases, and pesticide residue in agricultural and by-products are increasingly evident [1]. Natural and environmentally friendly pesticides of biological origin can effectively solve these problems. *Streptomyces*, a unique group of microorganisms, produce a diverse range of antibiotics, including peptide/glycopeptide, polyketide, tetracycline, phenolic, macrolide, anthraquinone, polyene, anthracycline, β-lactam, ansamycin, butenolide, benzoxazolinone, C17-glycoside, and lactone antibiotics [2]. These antibiotics not only play an irreplaceable role in the field of medicine but also hold great promise for applications in agriculture [3]. For example, avermectin and milbemycin are insecticides [4], and zhongshengmycin and kasugamycin are used to control leaf blight disease [5].

However, after years of research, isolating new natural products from common *Streptomyces* strains has proved challenging. The extreme *Streptomyces* populations found in extreme environments such as high altitude, high salinity, drought, and oceans may produce more natural products. The oceans are hypersaline in nature and most of the compounds produced by deep-sea actinomycetes have antimicrobial activity [6], and many novel compounds of important medicinal value have been isolated from marine actinomycetes, including salinosporamide A, salinipyrones A/B, iodopyridone produced by *Saccharomonospora* sp., and srenimycin produced by *Salinispora arenicola* [7]. Some thermotolerant actinomycetes produce heat shock metabolites at higher temperatures, and a new compound, murecholamide, from thermotolerant *Streptomyces* sp. AY2 was found to have inhibitory activity against cancer cell migration [8]. Pyridine-2,5-diacetamide is a newly discovered compound from actinomycetes in the Saudi Arabian desert with antimicrobial activity against *Enterococcus faecalis* and *Salmonella typhimurium* [9]. Between 2000 and 2021, antibacterial, anti-inflammatory, antiviral, antiallergic, antimicrobial, anticarcinogenic, and cytotoxic bioactivities have been identified from more than 50 new desert *Streptomyces* species [10].

Traditional isolation and purification strategies often struggle to achieve dereplication, which severely limits the discovery of novel unknown natural products. By combining genome mining with the antiSMASH program, the biosynthetic gene clusters (BGCs) responsible for the production of secondary metabolites in *Streptomyces* can be predicted to find secondary metabolites of interest. Using this method, it was predicted and found to produce kanamycin from *Micromonospora aurantiaca* 01 [11]. The predicted gene clusters can also be used to obtain target secondary metabolites by heterologous expression [12], but repeated trials are required due to the unpredictability of expression hosts and culture conditions [13]. It is worth noting that an unknown sequence forms a low confidence value with known sequences, while its BGC may encode structurally novel natural products with core structures different from known compounds. Conversely, if an unknown sequence forms a high confidence value with known sequences, its gene cluster may be closely related to known biosynthetic gene clusters, potentially producing structurally similar analogs with slight modifications [14].

The *Streptomyces* sp. KN37 was isolated from the high-altitude soil of the Kanas region in Xinjiang and identified as a special *Streptomyces* [15]. The fermentation broth of strain KN37 showed inhibitory effects against various plant pathogens, including *Rhizoctonia solani*, *Botrytis cinerea*, *Pseudomonas syringae*, *Alternaria solani*, *Sclerotinia sclerotiorum*, *Erwinia. amylovora* and so on [15]. In this study, based on genomics, the biosynthetic gene clusters of secondary metabolites were predicted. Combined with traditional isolation methods using activity-guided fractionation, secondary metabolites with strong antimicrobial activity were obtained from the fermentation broth of strain KN37. By employing modern identification techniques, the chemical structures of these metabolites can be elucidated, potentially laying the foundation for the development of new microbial-based pesticides.

## 2. Results

### 2.1. General Properties of the Streptomyces sp. KN37 Genome

The genome data of *Streptomyces* sp. KN37 were assembled using the Unicycler software (version: 0.4.8), distinguishing between the chromosomal and plasmid regions. The assembly resulted in one non-circular chromosome (Chr1) and two non-circular plasmids (Plas1, Plas2). Their respective lengths are 8,364,061 bp, 155,588 bp and 577,722 bp. The GC% content for each is 71.49%, 68.82% and 70.92%, respectively. The total length of the *Streptomyces* sp. KN37 is 9,097,371 base pairs (bp), with a total of 7965 predicted coding genes. The cumulative length of all the coding genes is 7,936,149 bp, accounting for 87.24% of the total genome length (Figure 1).

According to the results of the Kyoto Encyclopedia of Genes and Genomes (KEGG) annotation, most genes were annotated to Metabolism. In addition to the basic energy metabolism, some were annotated to ‘Metabolism of terpenoids and polyketides’ and ‘Biosynthesis of other secondary metabolites’, with 80 and 100, respectively, indicating that KN37 has certain secondary metabolite biosynthesis potential, especially terpenoids and polyketides. At the same time, there are up to 201 genes related to Membrane transport, which may enable KN37 to produce more extracellular substances (Figure 2).

### 2.2. Analysis of Secondary Metabolism Gene Clusters

Secondary metabolites are synthesized by microorganisms during specific growth stages, using primary metabolites as precursors. They do not have a defined function in the life activities of microorganisms and are not essential for growth and reproduction. However, secondary metabolites produced by *Streptomyces*, such as antibiotics, anticancer agents, anti-inflammatory agents, and enzymes, are important natural products that are commonly used in agriculture and human medicine [16]. Polyketide synthases (PKSs) can be classified into three types: Type I, also known as modular PKS, is a multifunctional enzyme complex composed of multiple domains. Type II, also known as aromatic PKS, is primarily involved in the synthesis of aromatic compounds. Type III, also known as chalcone PKS, is mainly responsible for the synthesis of monocyclic or bicyclic aromatic polyketides. The genome of *Streptomyces* sp. KN37 was analyzed using the antiSMASH program [17] (version 7.0.0), which identified 41 gene clusters associated with the biosynthesis of secondary metabolites. According to the predictions, these gene clusters are potentially involved in the production of carbohydrates, polyketides, terpenes, nonribosomal peptides (NRPs), and other substances. Several of these gene clusters have been annotated to known secondary metabolites, including antibiotics and bioactive compounds.

A total of 41 gene clusters were predicted, including 34 from Chr1 and 7 from Plas2. There were 13 gene clusters with more than 60% similarity (Table 1), among which the citrulassin E synthetic gene cluster, geosmin synthetic gene cluster, albaflavenone synthetic gene cluster, ishigamide synthetic gene cluster and ectoine synthetic gene cluster had 100% similarity. Some predicted high-similarity gene clusters that biosynthesize secondary metabolites are shown in Figure 3. The citrulassins contain N-terminal Leu and form a family with 55 members with lasso peptide structural features [18]. Ishigamide was once isolated from *Streptomyces* sp. MSC090213JE08 and was obtained by activating the silent gene cluster [19]. Ectoine is a widely distributed compatible solute accumulated by halophilic and halotolerant microorganisms to prevent osmotic stress in highly saline environments [20]. The biosynthetic pathway of pentalenolactone is related to the biosynthesis of sesquiterpenes, and its analogues 1-deoxy-8α-hydroxypentalenic acid and 1-deoxy-9β-hydroxy-11-oxopentalenic acid have certain antimicrobial activity [21]. Venturicidin is a class of macrolide antibiotics with antifungal activity against fungi such as *Venturia inaequalis*, *Sclerotinia fructigena*, and *Botrytis cinerea*, and it has potential agricultural uses [22]. Albaflavenone is a sesquiterpene antibiotic isolated from *Streptomyces*, which smells like soil camphor and has certain bactericidal activity [23]. Geosmin is the most common volatile substance in actinomycetes and is the source of the earthy smell [24]. Peucechelin is a macrolide antibiotic produced from *Streptomyces*, which has antibacterial effects on *Staphylococcus aureus*, *Micrococcus luteus*, *Sabmorella enteria* and *Proteus hauseri* [25]. Enterocin can inhibit *Listeria monocytogenes*, which is used to prevent contamination during cheese ripening [26]. Corbomycin is a glycopeptide antibiotic with a complex structure and has an inhibitory effect on *Staphylococcus aureus*. The mechanism of action is to prevent cell wall growth [27]. The Gausemycin family is a lipoglycopeptide antibiotic that has a certain antibacterial effect on Gram-positive bacteria [28]. Labyrinthopeptins are a class of carbacyclic lantibiotics. The most prominent representative is nisin, which has been known for its use as an antimicrobial food preservative for over 50 years [29]. These gene clusters with high similarity are closely related to known biosynthetic gene clusters and may synthesize similar secondary metabolites. Other low-similarity and unknown gene clusters may produce some novel secondary metabolites with different core structures from known natural products [14]. These new secondary metabolites have higher value in the study of natural products and may be able to be used as some new agents and drug synthesis precursors. In addition, the KN37 strain has excellent potential for secondary metabolite biosynthesis, which is worthy of further study.

### 2.3. Isolation of Active Substances by Traditional Methods

The KN37 fermentation broth was prepared, and the ethyl acetate crude extract was obtained from the supernatant of the fermentation broth. The crude extracts were separated by silica gel column chromatography, TLC and preparative liquid chromatography to obtain eight compounds. The chemical structures of these eight compounds (Figure 4) were determined by nuclear magnetic data (Appendix A). 4-(Diethylamino)salicylaldehyde is usually used in the synthesis of Schiff-base ligand by monocondensation with diaminomaleonitrile [30]. 4-Nitrosodiphenylamine is the rubber vulcanization accelerator. 2,4-Dimethylformanilide is a known environmental transformation product of Amitraz [31]. 4-Nitrocatechol is a known inhibitor of lipoxygenase [32]. Methylsuccinic acid has been reported to have a certain neuroprotective effect [33]. Phenyllactic acid has a wide range of antibacterial activity, which inhibits the growth of bacteria by destroying the cell wall or cell membrane of microorganisms [34]. 5,6-Dimethyl benzimidazole belongs to the class of organic compounds known as benzimidazoles. Ishigamide has been obtained from *Streptomyces* [19]. Aside from these eight compounds, except for Phenyllactic acid, no other compounds were found to have definite antibacterial activity.

Different from the predicted results, except for Ishigamide, we did not isolate the other predicted compounds, perhaps because the other compounds were not bacteriostatic or their gene clusters were not expressed, resulting in no production of these compounds. Therefore, the expression of these 13 high-similarity gene clusters was validated using RT-PCR. 

The core biosynthetic genes of gene clusters 1, 3, 4, 5, 6, and 7 were not expressed or not fully expressed, which means that the substances they predicted could not be synthesized or that intermediates and other substances were synthesized (Figure 5). The core biosynthetic genes of gene clusters 2, 8, 9, 10, 11, 12, and 13 were fully expressed, which to some extent confirms that these gene clusters are active. However, except for gene cluster 2, all the other biosynthetic genes in gene clusters 8, 9, 10, 11, 12, and 13 were partially unexpressed, especially genes 2647 and 2649 in gene cluster 8. It is hypothesized that they may have the ability to produce the predicted substance, but there is no guarantee that the structure of the substance produced is correct. By previous isolation, we obtained the compound Ishigamide produced by gene cluster 12. However, we did not obtain corbomycin and ectoine as predicted for gene clusters 2 and 10. This may be related to the content and the nature of the substances, which are difficult to obtain by conventional isolation methods when the content is too low. If these substances are soluble in water and difficult to dissolve in organic reagents, it is also difficult for us to extract them through ethyl acetate.

### 2.4. Preliminary Bioactivity Assays

The bioactivity of compounds 1~8 was determined. Compounds 1~7 showed different bioactivity against *R. solani* and *E. amylovora*. The content of compound 8 was too little, and the activity of compound 8 was not determined in this study. 

In the determination of antifungal activity against *R. solani* (Table 2), 4-(Diethylamino)salicylaldehyde demonstrated the highest antifungal activity, with an EC_50_ value of 14.487 mg/L, showing the strongest inhibitory effect. Next in line was 4-Nitrosodiphenylamine, with an EC_50_ value of 14.487 mg/L. In the determination of antibacterial activity against *E. amylovora* (Table 3), 4-Nitrosodiphenylamine showed the best bacteriostatic effect, followed by 4-Nitrocatechol, and their EC_50_ values were 5.715 mg/L and 19.871 mg/L. Overall, all seven compounds showed good inhibition of *E. amylovora*.

We also observed an interesting phenomenon. Compared with the control group, *R. solani* treated with N-(2,4-Dimethylphenyl)formamide of EC_50_ showed an increased transverse septa and became elongated, twisted and dried (Figure 6).

## 3. Discussion

As more and more actinomycetes with active substances are being isolated from cold, high-salt, deep-sea environments, attention is gradually turning to extreme environments. In recent years, genome-guided natural product prediction and isolation have together emerged as another commonly used method for natural product discovery. The advantage of this approach is that researchers can target and screen biosynthetic gene clusters (BGCs) of their interest for study, and to some extent, it is easier to discover new analogs. A number of new biologically active secondary metabolites have been identified through this approach. Examples include Spiroindimicin E and F [35], and two new α-pyrone derivatives, Amphichopyrone A and B [36].

*Streptomyces* sp. KN37 was isolated from the extreme environment of Kanas, Xinjiang [15]. We used genomics methods to analyze the whole genome of *Streptomyces* sp. KN37 and predicted its biosynthetic capacity. From the results, KN37 had good potential for secondary metabolite biosynthesis, especially terpenes and polyketides. The KEGG results also showed that KN37 had a good ability to synthesize extracellular substances, which was beneficial for us to obtain active substances from the supernatant of the fermentation broth. The results of the antiSMASH analysis showed that 41 biosynthetic gene clusters were predicted in KN37, and only 13 biosynthetic gene clusters had extremely high similarity. This shows that KN37 does have a high potential for secondary metabolite biosynthesis. However, after PCR verification, we found that most of the biosynthetic gene clusters were silent among the 13 highly similar gene clusters. In addition, there are still a large number of low-similarity gene clusters and silent gene clusters in KN37. The products of these clusters are unknown and worthy of further study.

As we know, genome mining is promising in terms of the discovery of new compounds, but there are still some challenges. Most gene clusters may remain silent under conventional culture conditions, indicating that the target compounds may not be obtained unless these gene clusters are activated [37]. Moreover, the prediction of BGCs is heavily reliant on existing databases, making it challenging to predict low-similarity or unexplored BGCs [12]. Although it is possible to predict the synthesis of natural products to some extent by predicting BGCs, it is not possible to determine whether unknown or new BGCs will synthesize the same or new natural products. As a result, genomics-guided natural product discovery methods have limitations and the variety of compounds discovered is very limited. Traditional separation methods have been widely used as natural product isolation techniques and have become quite mature. Researchers have successfully isolated numerous valuable new compounds with medicinal, food, and agricultural applications from both plant [38] and microbial [39] sources using traditional separation methods. A previously unreported stilbene derivative, pinosylvin monoacetate, and 14 known compounds were isolated from the leaves of *Nothofagus gunnii* using conventional isolation methods. Four known flavonoid natural products, catechin, quercetin, ayanin, and avicularin, were isolated from the leaves of *Nothofagus cunninghamii* [40]. Traditional separation methods are influenced by the compound abundance, making it easier to separate compounds present in higher quantities than those in lower quantities. There is also inherent randomness and uncertainty involved in these methods. Nevertheless, traditional separation methods can lead to the discovery of a diverse range of compounds.

We further explored the KN37 fermentation broth using traditional isolation methods. Eight compounds were obtained through isolation, purification and characterization: 4-(Diethylamino)salicylaldehyde, 4-Nitrosodiphenylamine, N-(2,4-Dimethylphenyl)formamide, 4-Nitrocatechol, Methylsuccinic acid, Phenyllactic acid, 5,6-Dimethylbenzimidazole and Ishigamide. These eight compounds are known but were first discovered in *Streptomycetes* sp. KN37. The results of the biological activity showed that compounds 1, 2, 3, 4, and 7 had good inhibitory effects on *R. solani* and *E. amylovora*, among which 4-(Diethylamino)salicylaldehyde and 4-Nitrosodiphenylamine had the best antimicrobial activity. After reading the literature, we found that except for Phenyllactic acid [34], the other compounds have not reported antimicrobial activity. Compared with the predicted results of the biosynthetic gene cluster, we obtained only one predicted compound, which proves that it is feasible to predict the synthesis of natural products by genomics. We also found the better inhibitory effects of 4-(Diethylamino)salicylaldehyde on *R*. *solani*, *Botrytis cinerea*, *Fusarium oxysporum*, *Corynespora cassiicola*, and *Sclerotinia sclerotiorum*. However, for reasons of space, the results are not presented in detail. The structure–activity relationship and mode of action of 4-(Diethylamino)salicylaldehyde are ongoing issues. The discovery of new compounds is becoming increasingly challenging, but there is still significant potential in studying the biological activities of known compounds. The known compounds Striguellone A, Isopanepoxydone, and Panepoxydone were isolated from *Lentinus strigellus*. They did exhibit different antibacterial activities against *Listeria monocytogenes*, *Enterococcus faecalis*, and *Pseudomonas aeruginosa*, which had not been reported before [41]. A sesquiterpene lactone named hymenoratin, which has never been associated with any biological activity, was isolated by biology-guided fractionation. This small molecule exhibited anticancer effects [42]. Hericene A and Hericenone F were found to have previously unreported anti-inflammatory activity [43]. Ungeremine from *Allium sativum* was found to have significant antibacterial activity against *Enterococcus faecalis* and *Pseudomonas aeruginosa* [44].

In the present study, no new secondary metabolites with activity could be identified from the KN37 fermentation broth. However, the bioactivities of these known compounds still have important research value. Next, these compounds may be used as chemical skeletons to synthesize some derivatives as new fungicides. We will also focus on silent gene clusters and low-similarity gene clusters, and we will continue to explore bioactive secondary metabolites that have not been found in *Streptomyces* sp. KN37.

## 4. Materials and Methods

### 4.1. Bacterial Strains and Cultures Conditions

The *Streptomyces* sp. KN37 (CGMCC No. 13160) was derived from a soil sample from the Kanas region in Xinjiang, which is permanently covered with snow, at an altitude higher than 3000 m above sea level and at a temperature lower than −5 °C [15]. The strain had been stored long-term in a culture medium containing 30% glycerol at a temperature of −80 °C. The *Streptomyces* sp. KN37 was deposited in the Department of Plant Protection, Faculty of Agriculture, Shihezi University. The following culture media were used: Gauze’s Synthetic Medium No. 1 (1 L water, 20 g soluble starch, 1 g KNO_3_, 0.5 g K_2_HPO_4_, 0.5 g MgSO_4_·7H_2_O, 0.5 g NaCl, 0.01 g FeSO_4_·7H_2_O, pH = 7.4–7.6, 18 g agar should be added for a solid culture) and Millet Medium (1 L water, 10 g millet, 10 g glucose, 3 g peptone, 2.5 g NaCl, 0.2 g CaCO_3_, pH = 7.2–7.4). The KN37 strain preserved at −80 °C in the freezer was inoculated on solid Gauze’s Synthetic Medium No. 1 and incubated at 28 °C for 7 days. A single colony was picked and streaked again for 7 days of cultivation. The activated strain was short-term preserved on agar plates. For liquid fermentation, 250 mL conical flasks were chosen. Eight 5 mm KN37 agar plugs were added to every 100 mL of liquid culture medium, and the mixture was incubated at 28 °C with agitation at 180 r/min for 3 days. The obtained KN37 seed liquid was then inoculated at a 4% ratio into 150 mL of culture medium and incubated at 28 °C with agitation at 160 r/min for 7 days, resulting in the KN37 fermentation broth.

### 4.2. Genome Analysis

The mycelium obtained from the Gauze’s Synthetic Medium No. 1 liquid culture medium was subjected to DNA extraction. In terms of the second-generation sequencing, the EXP-NBD104 kit from Oxford Nanopore Technologies and the NEBNext^®^ Ultra™ DNA Library Prep Kit for Illumina (NEB, Ipswich, Massachusetts, USA) were selected for the library construction and quality control on the Nanopore and Illumina platforms, respectively. The different libraries were sequenced on the Nanopore PromethION and Illumina NovaSeq PE150 platforms according to their effective concentration and target data volume. The raw data from each sample were analyzed using NanoPlot software (version: 1.29.1). The HiFi SMRTbell Libraries were prepared using the reagent Sequel II Sequencing Kit 2.0. The libraries were loaded into an SMRT Cell 8M Tray and sequenced on the PacBio Revio platform. The Unicycler software (version: 0.4.8) [45] was used for the genome assembly, combining both second- and third-generation sequencing data, and separating the chromosome and plasmid sequences. Gene prediction for the newly sequenced genome was performed using GeneMarkS software (version 4.17). The antiSMASH program [17] (version 7.0.0) was employed for genome annotation. The predicted protein sequences were compared against various functional databases using the Diamond alignment tool (evalue ≤ 1 × 10^−5^). For each sequence, the alignment with the highest score (default identity ≥ 40% and coverage ≥ 40%) was selected for annotation.

### 4.3. Nucleotide Sequence Accession Number

The whole-genome sequence of *Streptomyces* sp. KN37 has been uploaded to NCBI GenBank. Accession number: CP139044–CP139046. They are the accession numbers of Chr1, Plas1 and Plas2, respectively.

### 4.4. Isolation and Characterization of Secondary Metabolites

The *Streptomyces* sp. KN37 of *Streptomyces* was subjected to batch fermentation using Millet Medium, resulting in 80 L of fermentation broth. The fermentation broth was filtered and centrifuged at 4000 r/min for 30 min to remove precipitates. The supernatant was then subjected to consecutive extractions with ethyl acetate (1:1), followed by concentration under a vacuum at 50 °C, resulting in crude extract. The crude extract was subjected to crude separation using a silica gel column, yielding multiple fractions. Fractions with good biological activity were selected and further purified until the desired compound was obtained. The silica gel column was packed with 200–300 mesh silica gel powder (Shanghai Macklin Biochemical Co., Ltd., Shanghai, China). Thin-layer chromatography was performed using pre-coated silica gel GF254 plates (Qingdao Haiyang Chemical Co., Ltd., Qingdao, China), and the spots were visualized under UV light (254 nm) or using iodine staining. Alternatively, the silica gel plates were sprayed with 10% sulfuric acid-ethanol solution and heated for visualization.

Semi-preparative high-performance liquid chromatography was performed on a Shanghai Sinotac system (Sinotac Scientific Instruments Co., Ltd., Shanghai, China). The Nuclear Magnetic Resonance (NMR) spectra were recorded using a 400 MHz superconducting NMR spectrometer, AVANCE III HD (BRUKER, Fallanden, Switzerland), in either CDCl3 or DMSO-d6. 

### 4.5. Reverse Transcription PCR

Mycelium was obtained from the fermentation broth of the KN37 strain cultured for 7 d. The cDNA templates were obtained using the RNA extraction kit and the reverse transcription kit (Vazyme Biotech Co., Ltd., Nanjing, China). The RT-PCR reactions were performed separately according to the primers in Appendix A. At the end of the reaction, 5 μL of PCR reaction solution was taken for agarose gel electrophoresis, and the expression results were determined according to the brightness of the bands.

### 4.6. In Vitro Antimicrobial Activity Test

The phytopathogenic fungus *R. solani* was used as the test pathogen to evaluate the antifungal activity of the isolated compounds. The strain of *R. solani* was preserved in the Plant Protection Department of Shihezi University. The antifungal activity of the compounds against *R. solani* was assessed using a poisoned agar plate assay. Here, refer to the method of Yang et al. [46]. The strain was inoculated onto a potato dextrose agar medium containing the compounds, and after 3 days of cultivation at 28 °C, the inhibition rate of mycelial growth was recorded. The inhibition rate (%) was calculated using the following formula: Inhibition rate (%) = Wi/W × 100%, where Wi is the diameter of growth inhibition in the treated group, and W is the diameter of the blank control. The experiment was repeated three times, and the average value was calculated.

*E. amylovora* was preserved in the Plant Protection Department of Shihezi University. Refer to the method of Xiang et al. [47]. The agar plate dilution method was used to determine the inhibitory bacterial activity of the compounds and the bacterial inhibition rate was determined by counting. The inhibition rate (%) was calculated using the following formula: Inhibition rate (%) = Zi/Z × 100%, where Zi is the number of inhibitions in the treatment group and Z is the number in the blank control group. The experiment was repeated three times, and the average value was calculated. 

IBM SPSS Statistics software (version: 19.0.0) was used as a statistical analysis tool. The probit regression was used to calculate the virulence curve.

## Figures and Tables

**Figure 1 molecules-29-02040-f001:**
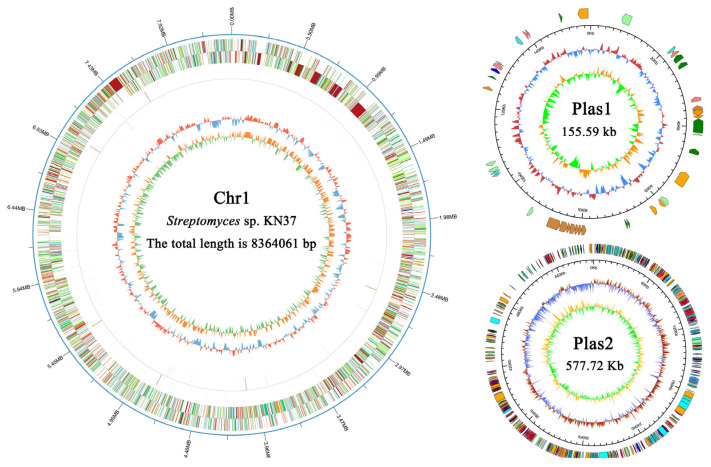
Whole-genome mapping. The outermost circle represents the genomic sequence position coordinates. From outer to inner, they respectively represent the gene functional annotation, ncRNA, and genomic GC content: The inward blue region indicates a lower GC content than the average GC content of the whole genome, while the outward red region indicates the opposite, with higher peaks indicating larger differences from the average GC content. Genomic GC skew value: The inward green region represents a lower content of G compared to C, while the outward orange region represents the opposite. The details of COG legend and ncRNA legend is shown in Appendix A.

**Figure 2 molecules-29-02040-f002:**
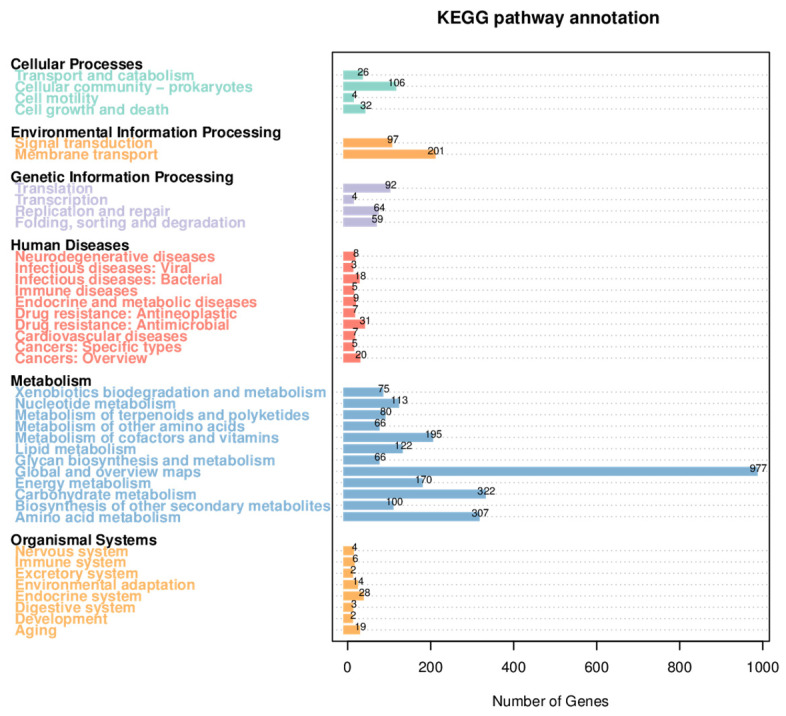
Distribution of the KEGG pathways in the *Streptomyces* sp. KN37 genome. By comparing with the KEGG database, the gene function annotation metabolic pathway classification map was obtained. The six main categories in the figure are green Cellular Processes, orange Environmental Information Processing, purple Genetic Information Processing, red Human Diseases, blue Metabolism and yellow Organismal Systems. The number on the bar graph represents the number of genes on the annotation.

**Figure 3 molecules-29-02040-f003:**
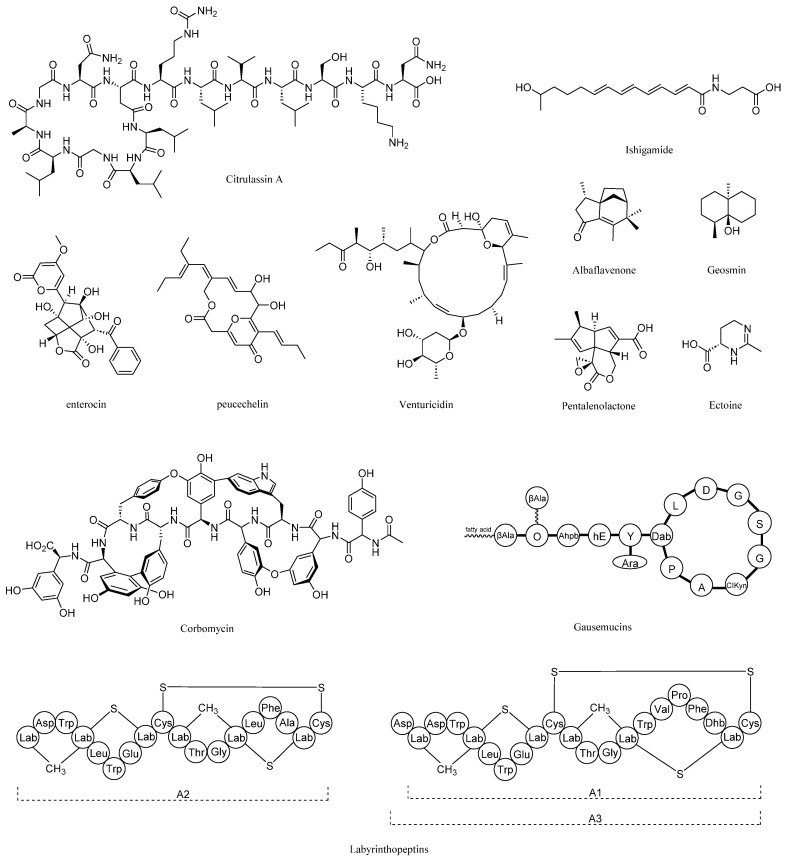
Some predicted high-similarity gene clusters biosynthesize secondary metabolites.

**Figure 4 molecules-29-02040-f004:**
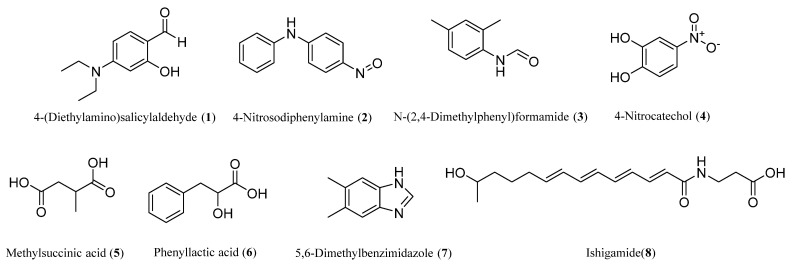
4-(Diethylamino)salicylaldehyde (**1**), 4-Nitrosodiphenylamine (**2**), N-(2,4-Dimethylphenyl)formamide (**3**), 4-Nitrocatechol (**4**), Methylsuccinic acid (**5**), Phenyllactic acid (**6**), 5,6-Dimethylbenzimidazole (**7**) and Ishigamide (**8**) isolated from the fermentation broth of *Streptomyces* sp. KN37.

**Figure 5 molecules-29-02040-f005:**
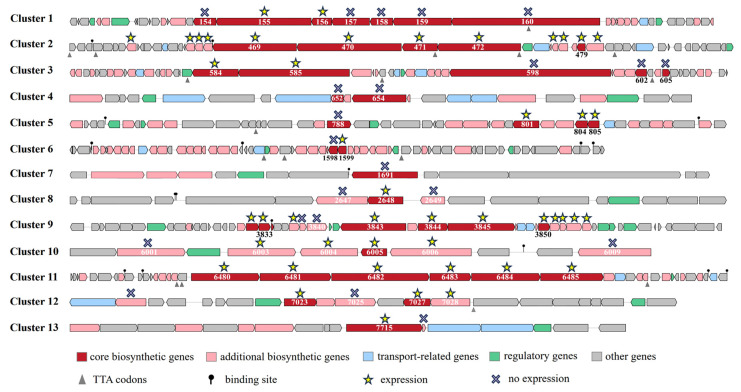
Expression of key genes in the biosynthetic gene clusters in *Streptomyces* sp. KN37.

**Figure 6 molecules-29-02040-f006:**
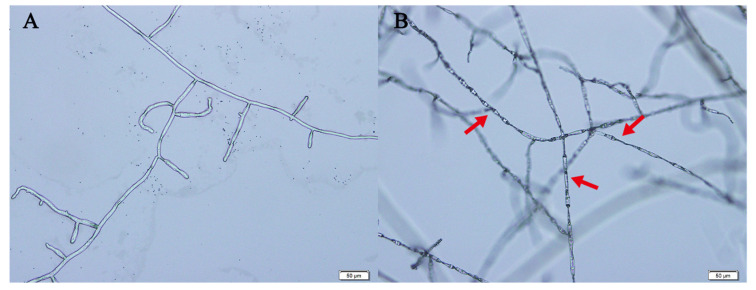
Changes in the mycelia of *R. solani* after treatment with N-(2,4-Dimethylphenyl)formamide. (**A**) Growth in 1% DMSO PDA medium. (**B**) Growth in EC_50_ N-(2,4-Dimethylphenyl)formamide PDA medium. Red arrows indicate mycelial malformations.

**Table 1 molecules-29-02040-t001:** Overview of the predicted secondary metabolites from biosynthetic gene clusters of the KN37 detected by antiSMASH.

Gene Cluster	Type	Size (kb)	Predicted Product	Most Similar Known Cluster	Similarity (%)	Reference Strain	Accession Number
1	NRPS-like transAT-PKS NRPS	103.717	NRP + Polyketide	oxalomycin B	75	*Streptomyces albus*	BGC0001106
2	NRPS T3PKS	88.789	NRP	corbomycin	96	*Streptomyces* sp. WAC 01529	BGC0002314
3	NRPS betalactone	104.296	NRP + Saccharide	gausemycin A, B	69	*Streptomyces kanamyceticus*	BGC0002430
4	lassopeptide	22.591	Ripp	citrulassin E	100	*Streptomyces glaucescens*	BGC0001551
5	NRPS-like terpene	50.493	Terpene	hopene	84	*Streptomyces coelicolor* A3(2)	BGC0000663
6	T2PKS	72.416	Polyketide: Type II polyketide	enterocin	90	*Streptomyces maritimus*	BGC0000220
7	terpene	21.082	Terpene	geosmin	100	*Streptomyces coelicolor* A3(2)	BGC0001181
8	terpene	17.556	terpene	albaflavenone	100	*Streptomyces coelicolor* A3(2)	BGC0000660
9	thioamide-NRP NRPS ladderane	73.74	NRP + Polyketide	ishigamide	100	*Streptomyces* sp. MSC090213JE08	BGC0001623
10	ectoine	10.404	Other: Ectoine	ectoine	100	*Streptomyces* sp.	BGC0002052
11	T1PKS	105.514	Polyketide	venturicidin D, E, F, A	76	*Streptomyces* sp. NRRL S-4	BGC0002454
12	terpene	25.638	Terpene	isorenieratene	85	*Streptomyces griseus* subsp. griseus NBRC 13350	BGC0000664
13	lanthipeptide-class-iii	22.582	RiPP: Lanthipeptide	labyrinthopeptin A1, A2, A3	60	*Actinomadura namibiensis*	BGC0000519

Note: Clusters 1 to 12 from Chr1, Cluster 13 from Plas2.

**Table 2 molecules-29-02040-t002:** Antimicrobial activity of 7 compounds against *R. solani*.

Compound	Toxicity Curve	*R* ^2^	EC_50_ (mg/L)	95% Confidence Interval (mg/L)
4-(Diethylamino)salicylaldehyde	*y* = 3.560*x* − 4.133	0.967	14.487	12.014–20.694
4-Nitrosodiphenylamine	*y* = 1.499*x* − 2.414	0.903	40.785	33.702–50.768
N-(2,4-Dimethylphenyl)formamide	*y* = 2.098*x* − 4.839	0.961	202.584	173.304–230.679
4-Nitrocatechol	*y* = 2.579*x* − 5.865	0.921	187.966	155.768–214.429
Methylsuccinic acid	*y* = 0.948*x* − 3.401	0.949	3868.586	2312.070–9543.943
Phenyllactic acid	*y* = 1.590*x* − 4.775	0.902	1009.024	833.982–1266.238
5,6-Dimethylbenzimidazole	*y* = 2.081*x* − 5.070	0.989	272.795	239.417–311.149

**Table 3 molecules-29-02040-t003:** Antimicrobial activity of 7 compounds against *E. amylovora*.

Compound	Toxicity Curve	*R* ^2^	EC_50_ (mg/L)	95% Confidence Interval (mg/L)
4-(Diethylamino)salicylaldehyde	*y* = 1.953*x* − 3.836	0.994	92.083	83.831–101.509
4-Nitrosodiphenylamine	*y* = 3.466*x* − 2.624	0.980	5.715	5.433–6.015
N-(2,4-Dimethylphenyl)formamide	*y* = 2.281*x* − 4.830	0.941	131.123	121.605–141.629
4-Nitrocatechol	*y* = 6.033*x* − 7.833	0.956	19.871	19.092–20.627
Methylsuccinic acid	*y* = 2.676*x* − 5.647	0.948	128.852	119.496–138.284
Phenyllactic acid	*y* = 3.415*x* − 8.977	0.899	424.891	400.864–450.311
5,6-Dimethylbenzimidazole	*y* = 4.768*x* − 8.960	0.968	75.716	72.100–79.502

## Data Availability

The original contributions presented in the study are included in the article/Appendix A; further inquiries can be directed to the corresponding authors.

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
