# Peer review of "Integrative Genomics and Bioactivity-Guided Isolation of Novel Antimicrobial Compounds from Streptomyces sp. KN37 in Agricultural Applications"

_molecules, 2024, doi:10.3390/molecules29092040_

Round 1
Reviewer 1 Report
Comments and Suggestions for Authors
The present study predicted and isolated antimicrobial bioactive compounds from Streptomyces using genome sequencing and traditional isolation and purification strategies which can have potentials as microbial-based pesticides. Overall, the work is significant, but there are several concerns in the manuscript.
Comments
1. Aren't these compounds targets beneficial microbes? Did you check whether the isolated compounds (4-(Diethyla- 20 mino)salicylaldehyde (1), 4-Nitrosodiphenylamine (2), N-(2,4-Dimethylphenyl)formamide (3), 4- 21Nitrocatechol (4), Methylsuccinic acid (5), Phenyllactic acid (6) and 5,6-Dimethylbenzimidazole (7)) have any negative impact on the beneficial plant microbes or is there any previous reports. Please write in discussion.
2. Line 58-70 and 77-79: How did genome mining with the antiSMASH program helps to identify and isolate the bioactive compounds in this present study?
3. What kind of sequencing technology is used in this work- Illumina, nanopore or PacBio? The manuscript says "both Illumina and nanopore" but NCBI CP139044.1 is PacBio. Please correct. Similarly check the assembly tools. Write the raw sequence and filtered sequence details that used for genome assembly.
4. Please add clear pictures about the antifungal and antibacterial activities against E. amylovora and R. solani, respectively.
5. Table 3: Please write the genomic regions of the predicted BGCs.
6. Line 196: Did you perform RT-PCR? write the methodology
7. Line 231: Please explain or remove Figure 6.
8. Please write examples of the successful application of Streptomyces derived or other natural compounds as microbial-based pesticides.
9. The discussion needs to be improved.
Other comments
Line 11: Its one of the main research direction.
Line 13: Please write more specifically "potent antimicrobial activity"
Line 16: Please add the BGC that helps in identifying the compounds.
Line 27: Please rewrite and explain how the newly predicted antibiotic-like substances still hold significant research value.
Line 73: Add reference for the statement "new species of Streptomyces"
Comments on the Quality of English Language
Minor editing of English language required
Reviewer 2 Report
Comments and Suggestions for Authors
Comments and Suggestions for Authors
This manuscript entitled Combining Bioinformatics Analysis and Traditional Separation to Obtain Antifungal Natural Products from Streptomyces sp. KN37, demonstrated potent antimicrobial activity of Streptomyces sp. KN37 through genomics and traditional isolation methods to obtain bioactive compounds from this strain. Although WGS was used in this study, no new secondary metabolites from the KN37 with activity could be identified, however, the identification of several obtained antibacterial compounds is very important for future investigations and research to develop new drugs. The manuscript is well written, and the material and methods are adequate; however, a small writing review would be beneficial. I recommend this manuscript for publication with minor revisions.
Results section:
Line 101: The first time a term is used do not use an abbreviation, such as “KEGG”.
Figure 2 is petite.
Material and Methods section:
This strain was obtained from soil or another type of sample?
Reviewer 3 Report
Comments and Suggestions for Authors
Upon reviewing the manuscript titled "Combining Bioinformatics Analysis and Traditional Separation to Obtain Antifungal Natural Products from Streptomyces sp. KN37" from a critical academic perspective, several key points and recommendations can be addressed:
The introduction adequately sets the stage by discussing the significance of actinomycetes and their secondary metabolites in addressing plant diseases and antibiotic resistance. However, the transition between general statements about the widespread use of pesticides and the specific focus on Streptomyces could be smoother. It is recommended that these topics be connected more explicitly by highlighting studies relating pesticide resistance to the need for novel antimicrobial compounds from sources like Streptomyces. The introduction section must be improved.
The methods section provides a detailed account of the genomic and traditional isolation procedures used. Nonetheless, it lacks critical evaluation of the potential biases these methods may introduce. For example, while whole-genome sequencing and KEGG enrichment are robust for predicting potential biosynthetic clusters, the reliance on known gene clusters may overlook novel biosynthetic pathways. It is advisable to discuss the limitations of these bioinformatic tools in potentially underestimating the metabolomic diversity of Streptomyces. There is insufficient evidence about the link between the genomic prediction and the found metabolites. I recommend including RNA differential expression assays for selected genes, which are part of the most representative clusters.
Some minor comments on this section are:
1. The authors should include the specific coordinates of the isolation place.
2. The strain should be deposited in a public collection.
The results present substantial data on the isolated compounds and their bioactivities. However, the discussion on the implications of these findings is somewhat superficial. The manuscript would benefit from a deeper analysis of how these compounds' activities compare to existing antimicrobials and their potential roles in agricultural settings. Specifically, addressing the mechanisms of action, possible resistance development, and comparisons with synthetic pesticides could enrich the discussion. There is a lack of information for linking the metabolite with the genes.
While the manuscript includes EC50 values and other quantitative data, there is a notable absence of statistical analysis concerning the reproducibility and significance of the findings. It would be prudent to include statistical tests to validate the conclusions drawn from the bioactivity assays.
The discussion section adequately highlights the study's contributions but could further elaborate on the novelty of the findings. The authors should clarify how the isolated compounds differ from those found in other Streptomyces species and the novel aspects of their biosynthesis pathways.
The manuscript uses current literature well, but some references to pivotal studies are outdated. Incorporating more recent studies could strengthen the narrative, especially in rapidly evolving fields like genomics and bioinformatics.
Comments on the Quality of English Language
English language use is generally reasonable. Moderate editing is required
Round 2
Reviewer 3 Report
Comments and Suggestions for Authors
Overall, the authors have responded thoughtfully and thoroughly to the comments and suggestions, demonstrating a commitment to improving the quality and impact of their research.
The authors have enriched the discussion section by including a deeper analysis of the compounds' activities, their mechanisms of action, resistance potential, and comparisons with existing antimicrobials. They also plan further studies to delve deeper into these aspects.
Finally, based on the content and focus detailed in the second version of the manuscript, I suggest that a more suitable title that encapsulates the primary findings and methods used in the study could be:
"Integrative Genomics and Bioactivity-Guided Isolation of Novel Antimicrobial Compounds from Streptomyces sp. KN37 in Agricultural Applications"
Comments on the Quality of English Language
Moderate editing of English language required
Author Response
Thank you very much for your review and valuable suggestions for our article. We appreciate your time and effort in helping us improve the quality and readability of our papers.
Based on your comments, we have revised the title of the artcle to "Integrative Genomics and Bioactivity-Guided Isolation of Novel Antimicrobial Compounds from Streptomyces sp. KN37 in Agricultural Applications".
Thank you again for reviewing our article and providing your valuable suggestions.